# Jasmonate ZIM Domain Protein (*JAZ*) Gene *SLJAZ15* Increases Resistance to *Orobanche aegyptiaca* in Tomato

**DOI:** 10.3390/plants13111493

**Published:** 2024-05-29

**Authors:** Siyu Chen, Lu Zhang, Qianqian Ma, Meixiu Chen, Xiaolei Cao, Sifeng Zhao, Xuekun Zhang

**Affiliations:** Key Laboratory at the Universities of Xinjiang Uygur Autonomous Region for Oasis Agricultural Pest Management and Plant Protection Resource Utilization, Agriculture College, Shihezi University, Shihezi 832003, China; 13199571099@163.com (S.C.); 13029617900@163.com (L.Z.); 15379722990@163.com (Q.M.); 13289938193@163.com (M.C.); tulanduocxl@sina.com (X.C.)

**Keywords:** *Orobanche aegyptiaca*, *Solanum lycopersicum*, *JAZ* family, VIGS, Heterologous overexpression, *SLJAZ15*

## Abstract

*Orobanche aegyptiaca* Pers. is a holoparasitic plant that severely reduces tomato (*Solanum lycopersicum* L.) production in China. However, there is a lack of effective control methods and few known sources of genetic resistance. In this study, we focused on key genes in the *JAZ* family, comparing the *JAZ* family in *Arabidopsis thaliana* (L. Heynh.) to the tomato genome. After identifying the *JAZ* family members in *S. lycopersicum*, we performed chromosomal localization and linear analysis with phylogenetic relationship analysis of the *JAZ* family. We also analyzed the gene structure of the *JAZ* gene family members in tomato and the homology of the *JAZ* genes among the different species to study their relatedness. The key genes for *O. aegyptiaca* resistance were identified using VIGS (virus-induced gene silencing), and the parasitization rate of silenced tomato plants against *O. aegyptiaca* increased by 47.23–91.13%. The genes were localized in the nucleus by subcellular localization. Heterologous overexpression in *A. thaliana* showed that the key gene had a strong effect on the parasitization process of *O. aegyptiaca*, and the overexpression of the key gene reduced the parasitization rate of *O. aegyptiaca* 1.69-fold. Finally, it was found that the *SLJAZ15* gene can positively regulate the hormone content in tomato plants and affect plant growth and development, further elucidating the function of this gene.

## 1. Introduction

Tomato (*Solanum lycopersicum* L.) is a widely grown vegetable crop around the world with many uses and high nutritional value. Broomrapes (*Phelipanche* spp. and *Orobanche* spp.) are parasitic plants that affect the growth of cash crops throughout the world, causing major parasitism of tomato, oilseed rape (*Brassica napus* L.), legumes (*Fabaceae* spp.), and sunflower (*Helianthus annuus* L.). They parasitize on plant roots, which can lead to devastating yield losses and is an increasing threat to tomato cultivation. A wide range of control methods are available for parasitic weeds, but no effective control strategy has been developed to date [1]. Traditional control methods include seed quarantine, trapping, manual removal, induced eradication, and chemical control [2]. Current research suggests that breeding resistant species is also an effective strategy and much work has been conducted to identify resistance (R) genes; however, the function of the R genes may be overcome by new strains of parasitic plants, as most R genes mediate race-specific resistance through a gene-for-gene mechanism [3,4,5].

Currently, modified plant viral vectors are used to express RNAi inducers, which is known as virus-induced gene silencing (VIGS). VIGS has many advantages. First, it can be applied to a wide range of host plant types. Some virus vectors can infect monocotyledons, legumes, and even perennial woody plants, for which *Agrobacterium tumefaciens*-mediated transformation is difficult. Second, viruses can achieve systemic infection of plants [6] The selection of ideal target genes, especially those involved in the parasitization process, is crucial for VIGS to control parasitized plants. *Agrobacterium*-mediated VIGS was first used in the *Nicotiana benthamiana* experiment, and researchers have successfully used it in *Solanaceae*, *Lactuca sativa*, *Arabidopsis thaliana*, and *S. lycopersicum* by optimizing it and obtained stable silencing results [7,8]. Because VIGS can specifically induce the silencing of genes in various parts of the plant, it is often used as an efficient reverse genetic technique in the study of functional genes in the plant body, especially in the study of the function of certain disease resistance genes and disease resistance signaling pathways, which plays an increasingly important role. Tobacco rattle virus (TRV) was originally discovered in tobacco and was shown to be able to replicate, translocate, and efficiently transmit silencing signals to plant tissues in host plants, opening up the possibility of studying gene function in primary and secondary plant metabolism [9]. In recent years, the use of TRV as a vector to induce transient silencing in plants has been rapidly developed, not only in tomato and tobacco but also in other lycopene plants. Ghamsari successfully induced silencing of the PDS gene in *Datura stramonium* using the TRV vector system [10], and Zhang successfully induced silencing of the PDS, MPE2, and MPF3 genes in *Clinopodium polycephalum* with high efficacy [11]. In addition, TRV vectors can often be used to introduce the host’s PDS gene into the vector along with it so that it can be used as a reporter gene for silencing through the phenomenon of photobleaching. Phytoene desaturase (PDS) is a key enzyme in the plant carotenoid synthesis pathway and is located upstream of the carotenoid synthesis pathway, which catalyzes the conversion of colorless octahydro lycopene into colored carotenoids [12]. Carotenoids are important pigments that constitute the color of different haustorium in plants and are mainly distributed in the pigment cells of plant haustorium, which make plant haustorium, such as flowers and leaves, show different colors [13]. When the PDS gene is silenced, the plant carotenoid synthesis pathway is blocked, which leads to the breakdown of chlorophyll and causes albinism and dwarfism in plants [14]. Recently, it has been demonstrated that jasmonic acid (JA), salicylic acid (SA), and ethylene glycol all play important roles in host resistance to parasitic plants [15]. In the study of JA, we found that it plays a crucial role in host resistance re-response to parasitic plants. For example, JA and phytochemical synthesis-related genes are highly expressed in *Lotus japonicus* (Regel) K. Larsen when it is parasitized by *O. aegyptiaca* and *S. hermonthica* [16]. The knockdown of a JA synthesis gene in *S. hermonthica*-resistant rice resulted in a substantial increase in susceptibility, while the application of exogenous JA restored the resistant phenotype [17]. In addition, JA promotes the thickening of plant cell walls, thereby directly or indirectly resisting pathogen invasion [18].

The JA signaling pathway is a more complex process, which includes a large number of related synthetic genes and proteins. Through searching the literature, we found that the JA signaling pathway mainly includes downstream gene response, JA signaling, and the creation and metabolism of signal transduction molecules [19]. In response to external biotic or abiotic stresses, plants produce substantial amounts of jasmonic acid, which is subsequently converted into the downstream compound JA-Ile by the adenylate-forming enzyme *JAR1* [20]. Subsequently, JA-Ile further interacts with the jasmonate receptor F-box protein *COI1* (*Coronatine Insensitive* 1) to generate the JAS structural domain. Additionally, *JAZ* is believed to serve an inhibitory function in the JA pathway [21]. When JA-Ile is absent from the pathway, JAZ proteins collaborate with NINJA proteins [22] to integrate the co-inhibitory factor, TPL (topless). This interaction allows JAZ proteins to synergize with downstream transcription factors, such as MYC2, preventing MYC2 from transcriptionally activating JA-responsive genes. When JA-Ile is present, JA-Ile binds to COI1 to form the COI1-JAZ complex. Upon formation of these complexes, JAZ proteins are ubiquitinated by the E3 ubiquitin ligase SCFCOI1 (Skp/Cullin/F-box) complex and used to interpret JAZ repressors [23]. Among these, MYC2 can work in concert with MTB (MYC2-targeted HLH) to modulate jasmonic acid signaling [24]. Numerous members of the JAZ gene family, each with a distinct biological role, are involved in the control of plant development, responses to abiotic stressors, and abiotic stress tolerance. For example, the overexpression of the *OsJAZ9* gene improves rice (*Oryza sativa*) tolerance to potassium deficiency by changing the JA level and JA signal transduction pathway [25]. The overexpression of the *GsJAZ2* gene in soybean (*Glycine max*) significantly enhanced the resistance of transgenic lines to saline stress [26] The overexpression of *AtJAZ1* in *Arabidopsis* can enhance host resistance to *Spodoptera exigua* [27] The overexpression of *OsJAZ*s in rice can lead to malformations in haustorium development [28,29]. Previous studies showed that most *SLJAZs* are induced by JA and are able to respond to a wide range of biotic and abiotic stresses, which provides clues to analyze the function of JAZ genes in biotic and abiotic responses in tomato [30,31].

In previous studies of the JAZ family of genes in tomato, we have identified many genes that are resistant to biotic stresses. For example, *SLJAZ2* (Solyc12g009220), *SLJAZ6* (Solyc01g005440), and *SLJAZ7* (Solyc11g011030) silenced tomato plants and showed enhanced disease-associated cell death for *Pst* DC3000 (*Pesudomonas syringae* pv. Tomato DC3000) [31]. In addition, *SLJAZ25* (Solyc12g009220) interacted with *SLJAR1* (Solyc07g042170), *SLCOI1*, *SLMYC2*, and other resistance-related genes to form a regulatory network, and these genes played an important role in the regulation of tomato gray leaf spots [32]. Moreover, it has been reported that *SLJAZs* are involved in tomato plant development, and *SLJAZ2* (Solyc12g009220) is involved in mediating the transition from early asexual to reproductive growth in plants [33]. From the above studies, it was found that despite the different nomenclature, the Solyc12g009220 gene has a pivotal position in the JAZ family. It plays a role in the development of plants towards sexual regeneration, as well as in the development of resistance to biotic stresses.

By screening the whole genome of tomato, we hoped to obtain all the genes of the *JAZ* family in tomato. The screened genes were also structurally analyzed. Finally, we aimed to achieve effective silencing of the tomato host plant using TRV (tobacco rattle virus) and identify the level of resistance of the target gene to *O. aegyptiaca* using VIGS technology. Then, we accurately located the action position of the resistance gene using subcellular localization technology to lay the foundation for further resolving its function. Finally, the mechanism of action was further elucidated through the determination of heterologous overexpression in *A. thaliana* and the change in enzyme activity after parasitism by *O. aegyptiaca*.

## 2. Results

### 2.1. Identification and Characterization of JAZ Genes in Tomato

After visualizing the data using several pieces of software, such as HMMER (hidden Markov model of contours), the Simple Module Architecture Research Tool (SMART), the Conserved Structural Domains Database (CDD), and TBtools, the results revealed that 16 *SLJAZ* genes were localized on 12 chromosomes of tomato, and *SLJAZ* genes were present on chromosomes 1, 3, 6, 7, 8, 9, 11, and 12. One to three *SLJAZ* genes were present on the above chromosomes, with only one *SLJAZ* gene on chromosomes 7, 9, and 11; two on chromosomes 3, 6, and 12; and three on chromosome 1. These genes were named *SLJAZ1* to *SLJAZ16* for the convenience of subsequent studies, and their locations on the chromosomes are shown in Figure 1A. Gene duplication is an important factor in gene functional differentiation and gene amplification [34]. The results show that we detected five pairs of segmental duplications in *SLJAZ* genes, such as *SLJAZ4* and *SLJAZ6*, *SLJAZ5* and *SLJAZ16*, *SLJAZ7* and *SLJAZ13*, and *SLJAZ8* and *SLJAZ15* (Figure 1B).

### 2.2. Phylogenetic Relationships and Gene Structure Analysis of SLJAZ Genes

To further clarify the phylogeny of *JAZs* in tomato, a phylogenetic tree of 16 SLJAZ proteins and 23 AtJAZ proteins expressed in *A. thaliana* was constructed using molecular evolutionary genetics analysis (MEGA X) (Figure 2A). As shown in the figure, the thirty-nine *JAZ* genes were categorized into five subfamilies, of which only four branches contained both screened *JAZ* genes from tomato and *A. thaliana*. One of our most interesting genes, *SLJAZ15*, was located in subfamily III, with the strongest homology to *SLJAZ8* (Figure 2A,B).

Based on the above results, we divided the phylogenetic tree into four branches (Figure 2B). After analyzing the genetic structure and visualizing the data, we found that most of the *SLJAZ* genes consisted of four exons and three introns (Figure 2), and the rest of the *SLJAZ* genes contained more than five exons, among which *SLJAZ2* had the highest number of exons (i.e., 15), and *SLJAZ15* and *SLJAZ8* had six exons. The above conditions indicate that the two genes have a high probability of homology. To further explore the possible functions of *SLJAZ* genes, we analyzed the conserved motifs of the nine proteins and found that all members contained the Jas motif (Motif 5) (Figure 2). In the phylogenetic tree, five other motifs were specific to different branches. The functional differences in the tomato *SLJAZ* genes may be due to the clade-specific distribution of conserved motifs.

### 2.3. Resistant Plants Showed Increased Parasitism of O. aegyptiaca after Silencing SLJAZ15

In tomato plants infected with *Agrobacterium tumefaciens* carrying the TRV-PDSi construct, significant bleaching was observed in the leaves after 15–20 d of incubation, and subsequent experiments were performed after stabilization of the bleaching phenomenon (Figure 3A). To further determine the efficiency of VIGS, quantitative real-time polymerase chain reaction (qRT-PCR) assays were performed on leaves collected from plants containing the TRV2:00i empty vector and the TRV2:*SLJAZ15*i ced plants compared with that in the TRV2:00i empty vector-infected plants (Figure 3B). These results collectively suggest that the target gene was successfully knocked down in resistant plants. A comparison of the root chamber method with the potting method revealed that *SLJAZ15*-silenced plants showed symptoms of susceptibility, i.e., an increase in the rate of *O. aegyptiaca* parasitism by 6.32 and 4.84 flod, respectively, of the mean original, whereas the untreated control plants showed better resistance characteristics to *O. aegyptiaca*. As shown in Figure 3C,D, after 20 days of co-cultivation of different treated plants with *O. aegyptiaca*, the average parasitization rate of a single plant was significantly increased in plants silenced with *SlJAZ15* compared to plants carrying the TRV2 vector alone. Growth was increased by 47.23%, 51.39%, and 91.13% at different periods of S4, S5, and S6, respectively. Based on the potting method of co-cultivation (Figure 3E,F), after silencing the resistance genes, the plants produced obvious deformities and dwarfing, accompanied by the problem of impeded growth of leaves. Additionally, using the root chamber method for secondary validation, we found that co-cultivation produced the same pattern in the growth of *O. aegyptiaca*. As shown in Figure 3, the root growth of the plants was enhanced after co-cultivation, and the parasitism of *O. aegyptiaca* was more significant. An analysis showed that the three phases had increased 1.86-, 5.59-, and 7.09-fold, and in the co-cultivation phase of the root chamber, the developmental attitude was best in the period of S5, which was significantly different from that in the TRV2:00i plants. According to the longitudinal comparison experiments, *O. aegyptiaca*-parasitized TRV2:00i plants were more prone to necrosis and less likely to be parasitized by *O. aegyptiaca*, while *O. aegyptiaca*-parasitized TRV2:*SLJAZ15*i plants had increased root growth, grew more vigorously, and had a good developmental status.

### 2.4. Subcellular Localization of SLJAZ15 in Nicotiana tabacum

To determine where *SLJAZ15* exerts resistance in plants, we utilized a GFP fusion protein expression assay to determine where the *SLJAZ15* gene is expressed in cells. As shown in Figure 4, pCAMBIA2300-*GFP* can be clearly seen under the microscope with obvious fluorescence signals in the cell membrane and nucleus of the lower epidermal cells, but *SLJAZ15-GFP* only has a fluorescence response in the nucleus, which confirms that *SLJAZ15* is localized in the nucleus.

### 2.5. SLJAZ15 Overexpression in A. thaliana Increased Resistance to O. aegyptiaca

To investigate whether the function of *SLJAZ15* is conserved, the *A. thaliana* Floral Dip method was used to obtain positive *Arabidopsis* plants overexpressing *SLJAZ15.* In the root chamber method, after co-cultivating *O. aegyptiaca* and transgenic *A. thaliana* for 20 d, the number of *O. aegyptiaca* parasitizing transgenic plants was significantly reduced compared with that of the control *Arabidopsis*, with an average 1.5-fold reduction in the number of parasitizing *O. aegyptiaca* per plant (Figure 5D). The average number of *O. aegyptiaca* parasitizing WT plants was 68.4 at different times, while the number of *O. aegyptiaca* parasitizing transgenic plants was only 40.4 at different times, which was a 69% reduction (Figure 5E,F). Moreover, *O. aegyptiaca* parasitizing overexpressing plants browned and died at a later stage, and their developmental status was significantly worse than that of *O. aegyptiaca* parasitizing common *A. thaliana*. Statistical findings from the growth and development of the host *A. thaliana* showed that the plant height and the growth and development of transgenic plants were significantly better than those of the WT after assessing *O. aegyptiaca* using the potting method (Figure 5A,B).

### 2.6. Physiological and Biochemical Analyses

JA is an important hormone that regulates plant growth and development, and it plays a very important role in the plant body. *SLJAZ15* is one of the genes of the *JAZ* family involved in the regulation of plant growth and development, as well as in the defense against biotic and abiotic stressors. Therefore, to further analyze the role of this gene, we measured the JA content of tomato plants and analyzed the function of the gene according to changes in content. Based on the measured data (Figure 6), we observed that the JA content in normal tomato plants was significantly higher than that in *SLJAZ15*-silenced plants, with 3.72 and 3.24 levels, respectively. Both normal and *SLJAZ15*-silenced transgenic plants showed a certain increase in JA content after 20 d of parasitization with *O. aegyptiaca*, but the increase was small.

According to these experiments, with either VIGS or overexpression, the height and growth status of transgenic *SLJAZ15* plants were altered to different degrees. In the *SLJAZ15* silencing system, the silenced plants grew poorly, became shorter in height, and even became deformed; however, in the overexpression system, the growth of *A. thaliana* accelerated to different degrees.

## 3. Discussion

Tomato (*S. lycopersicum*) is widely grown all over the world and is a very important cash crop. However, during its growth and development, it is susceptible to a variety of pests and diseases, which can lead to a decrease in yield. From previous studies, we know that plants have evolved very complex defense mechanisms [35] in the process of developing continuous resistance to biotic stress, including programmed cell death, allergic necrosis, secretion of antimicrobial substances, and constant changes in endogenous hormones [36]. At the same time, the evolution of the genome of a plant is particularly important for it to better adapt to environmental changes [37]. Individual genomes play their own roles and functions, thus forming a network of disease [38,39] resistance results, which is a hotspot of forthcoming genetic research.

Jasmonate zinc finger proteins expressed in inflorescence meristem (ZIM) domain proteins (*JAZ*) are important negative regulators within the JA signaling pathway and play an important role in the plant defense response [40]. When plants are subjected to biotic or abiotic stresses, the natural signaling molecule JA can activate the expression of downstream response genes and give them the ability to cope with the stress [41]. Furthermore, JA participates in plant growth and development and defense responses, such as pathogen infection and insect injury [42,43,44]. The JA content is dynamically balanced in plants. When the JA content in plants is low, JAZ proteins bind to *MYC2* transcription factors (TFs) to inhibit the transcription of downstream response genes [45]. In contrast, JAZ proteins bind JA (-Ile) receptor complexes for subsequent degradation by the 26S proteasome, enabling the transcriptional activity of MYC2 TFs and ultimately activating the JA signaling pathway [46]. Twelve members of the *JAZ* gene family have been identified in *Arabidopsis thaliana* [47]. In addition, they have been reported in rice, maize, soybean, and tobacco crop species [48,49,50]. In tomato, some attention has been devoted to the identification, characterization, and functional analysis of *JAZ* gene family members [30,31,51].

### 3.1. Analysis of the JAZ Gene Family in Tomato

The *JAZ* gene family is present in a wide range of plants. There are 23 *JAZs* in *A. thaliana* [51], 15 in rice [52], 16 in maize [48], 36 in turnip [53], 30 in cotton [54], and 11 in grape [55]. We found that *SLJAZ5* was located on chromosome 3; *SLJAZ6* and *SLJAZ7* on chromosome 6; *SLJAZ8* on chromosome 7; *SLJAZ9*, *SLJAZ10*, *SLJAZ11*, and *SLJAZ12* on chromosome 8; *SLJAZ13* on chromosome 9; *SLJAZ14* on chromosome 11; and *SLJAZ15* and *SLJAZ16* on chromosome 12. We grouped these sixteen *JAZ* genes into five groups according to their grouping pattern in the *JAZ* gene family in *Arabidopsis thaliana*, and the genes in the same branch could have similar or complementary physiological functions.

According to the results of previous studies, the structures of introns and exons represent different biological functions of genes, and genes with the same structure of identical introns and exons may have the same gene function [56]. In previous studies, *JAZ* family members were divided into five subfamilies, and genes on the same branch have similar structures and functions [57]. In the present study, we divided the sixteen *SLJAZ* genes into four subfamilies. Most of the *SLJAZ* genes contained different numbers of introns and exons, and only the *SLJAZ2* gene did not contain introns, suggesting that introns have been deleted in these genes, which led us to note that the precise loss and gain of introns may be an important factor in promoting de novo gene birth [58]. Based on gene theme analysis, we aimed to find nine themes in sixteen *SLJAZ* genes. Among them, we found the shared Jas motif, which is contained in all *SLJAZ* genes. Within the same clade, some motifs are unique and form the basis of gene family classification and functional differentiation.

Plant evolution and gene family expansion require the advancement of gene duplicate expression [57]. Studies have shown that 70–80% of angiosperms experience gene duplication or polyploidy events [59]. Effective means of generating new genes and developing resistance to foreign invaders would include methods such as gene fragment duplication, base shifting, and replication, as mentioned previously [60]. In this study, we found that there were four pairs of duplicated genes in the *SLJAZ* genes we screened. To continue exploring the phylogenetic relationships between the *SLJAZ* genes and the *SLJAZ* genes of other plant species, we then determined the co-lineage relationships between tomato and *A. thaliana*. Finally, eight pairs of co-linear *JAZ* genes were identified between tomato and *A. thaliana*.

### 3.2. Functional Analysis of the SLJAZ15 Gene

To gain a more detailed understanding of the resistance of genes in this gene family to *O. aegyptiaca*, we screened the key gene *SLJAZ15*, which is 756 bp across the field and located on chromosome 12. The resistant tomato variety “H1015” appeared to be weakened by the phenomenon of reduced resistance, and the rate of parasitization increased by 47.23–91.13%; along with the increase in the rate of parasitization, the plants appeared to be short and misshapen. This demonstrates that *SLJAZ15* is in a critical position for plant resistance to biotic stress, and silencing this gene affects the ability of plants to resist parasitism by *O. aegyptiaca*. Moreover, impaired plant growth and development demonstrated that this gene is related to plant growth and development.

To further understand the function of the *SLJAZ15* gene family, we analyzed it using subcellular localization, which, as predicted, showed that *SLJAZ15* was localized in the nucleus. This is consistent with the expectation that the *SLJAZ15* gene is involved in the plant resistance stress response.

Next, we obtained *A. thaliana* plants heterologously overexpressing *SLJAZ15*. The results were in line with our expectation that *Arabidopsis* plants overexpressing the *SLJAZ15* gene would grow better, with better stem and leaf development, as well as enhanced *O. aegyptiaca* resistance, with a significant reduction in *O. aegyptiaca* parasitism to approximately 59.06% of the WT. Moreover, transgenic *A. thaliana* root development was better in the absence of *O. aegyptiaca* inoculation, with a longer root length and larger leaf development.

### 3.3. Physiological and Biochemical Analyses of SLJAZ15

Physiological and biochemical analyses are an important way of revealing the interactions of various substances and molecules in an organism, and analyzing the content, structure, and function of various types of substances in an organism allows us to understand its metabolic processes and the differences between organisms in different states. In this study, we found that the JA content in transgenic plants after *SLJAZ15* silencing was significantly reduced. Side-by-side comparisons also revealed that after parasitizing *O. aegyptiaca*, both *SLJAZ15*-silenced plants and control plants showed a significant increase in JA content in vivo, suggesting, in agreement with our results, that JA has a positive role in defense against external biotic stress.

Interestingly, we also found a role for JA in the plant growth state at the same time. In previous studies, we found that large amounts of exogenous JA cause stunted plant growth [61], while moderate amounts promote plant growth [62]. Even in the *JAZ* family, there are important genes that can regulate the morphology of the plant and accelerate its flowering [33]. In this experiment, the key gene, *SLJAZ15,* was silenced, and the plants were dwarfed. The dwarfing phenomenon was more significant after parasitism by *O. aegyptiaca*, and the silenced plants were more prone to aberrant growth.

### 3.4. Conclusions

*O.aegyptiaca* is a holoparasitic plant. It mainly parasitizes plant roots and severely reduces the yield of tomato in China. However, conventional control methods are ineffective, and there are few known sources of resistance genes. As an important part of the JA signal transduction pathway, *JAZ*s play a prominent role in plant disease resistance. First, we identified 16 *SLJAZ* genes in tomato. These genes clustered into four branches, and their gene structure and motif distribution in the same group were similar. Based on previous research, the promoter regions of these genes might contain many cis-acting elements associated with disease resistance. We subsequently confirmed the negative regulatory effect of the *SLJAZ15* gene on tomato resistance to *O.aegyptiaca*. Additionally, we also observed that following the allogeneic overexpression of the *SLJAZ15* gene, *A. thaliana*’s growth and development were notably improved, the parasite load was significantly reduced, and the nodules were susceptible to necrosis. Lastly, we also confirmed that the *SLJAZ15* gene plays a significant role in the regulation of hormone levels in plants and the subsequent impact on plant growth. These results not only show the importance of *SLJAZ*s in tomato disease resistance but also lay a good foundation for further study of the potential regulatory mode of *SLJAZ* genes.

## 4. Materials and Methods

### 4.1. Identification of SLJAZ

The hidden Markov model of contours (HMMER) (http://www.hmmer.org/, accessed on 23 December 2023) of known *JAZ* structural domains was downloaded from the Protein Family of Structural Domains (Pfam) database (http://pfam.xfam.org/, accessed on 23 December 2023). Tomato genome and *A. thaliana* genome sequence data were obtained from the NCBI database (https://www.ncbi.nlm.nih.gov, accessed on 24 December 2023). To eliminate pseudogenes, the REtrotransposed Gene EXPlorer was used for filtration. Structural domains were validated using the Simple Module Architecture Research Tool (SMART) (http://smart.embl-heidelberg.de/, accessed on 24 December 2023) and the Conserved Structural Domains Database (CDD) (https://www.ncbi.nlm.nih.gov/cdd/, accessed on 24 December 2023) online tools; the data were visualized using TBtools.

### 4.2. Chromosomal Mapping, Phylogenetic Analysis, and Gene Sequence Analysis of JAZ Gene Family Members in Tomato

The chromosome positions of *SLJAZ* genes were retrieved using the Solanaceae Genomics Network (https://solgenomics.net/, accessed on 28 December 2023), and the chromosomal distribution of *SLJAZ* genes was mapped using TBtools. The phylogenetic trees of Arabidopsis thaliana and tomato *JAZ* sequences were then constructed after multiple sequence alignment using the MEGA X software ClustalW (default parameters), and finally, the constructed evolutionary trees were optimized using the Evolution View online tool (https://www.evolgenius.info/evolview/, accessed on 28 December 2023). Protein sequences were analyzed using the Multi EM for Motif Elicitation online program (http:/mememe.nbcr.net/meme/intro.html, accessed on 29 December 2023), with the analysis parameters set to the default values and Motif set to 9.

### 4.3. Plant Material and Treatments

“Heng Shi 1015” is a processed tomato variety that has proven resistance to *O. aegyptiaca* in our laboratory. The tomato variety H1015 was planted as experimental material at the experimental station at Shihezi University. A 16/8 h light/dark cycle, 35,000 lx of light intensity, 45% relative humidity, and a diurnal temperature range of 20–25 °C were all applied to the growth of the seedlings. The tomato seedlings were injected with bacterial suspensions mediated by *Agrobacterium* (TRV-*SLJAZ15*) after they had developed two true leaves. Following therapy, qRT-PCR assays were carried out. Ten seedlings were selected for each treatment, and all treatments were repeated three times. Leaves of *SLJAZ15*-silenced plants were collected for the subsequent determination of physiological indices, immediately frozen in liquid nitrogen, and stored at −80 °C.

*Arabidopsis* plants were obtained by *Agrobacterium*-mediated floral dipping. Independent transgenic lines were identified by evaluating kanamycin resistance. The T2 homozygous progenies of transgenic lines were selected for further study. *Arabidopsis* seeds of the wild type (WT) and transgenic *SLJAZ15* were surface sterilized for seven minutes using 30% NaClO and eight minutes using 70% ethanol. The seeds were then washed three times with distilled water. Twenty-one-day-old plants growing in a large culture plate were irrigated with Hoagland nutritional solution in order to analyze resistance and determine physiological indices in mature plants. Additionally, the parasitic ability of *O. aegyptiaca* was examined. The *Arabidopsis* plants, both WT and transgenic *SLJAZ15*, were cultivated in a plant culture chamber with long days (16 h of white light, 8 h of darkness, and 60% relative humidity at 21 °C).

The mass ratio of tomato seeds to culture soil (nutrient soil: vermiculite: 2:1) for the potting procedure was 2000:1. Plants (20 cm height and 15 cm diameter) containing the soil mixture were used for their planting, and they were kept in an artificial climate room (20–28 °C, 40–60%, 16/8 h, 10,000 lx). In this investigation, three duplicates of each species were put up. After 25 days, root wash surveys were carried out to tally the number of nodules and *O. aegyptiaca* that emerged from each pot. Two biological duplicates were set up with thirty treatments and ten controls. The interaction between *O. aegyptiaca* and the tomato root system was simulated using the root chamber approach. Initially, the tomato seeds were sterilized with 5% NaClO for 20 min at 160 rpm and 25 °C in a shaker. The shaker then continued to shake, and the germinating seeds were incubated at room temperature. After a period of elongation and development of the canals, they were transferred to a glass Petri dish with a sterile sponge covered with two layers of sterile filter paper (15 cm in diameter) and incubated until new roots grew. When the new roots had grown to approximately 2/3 of the volume of the Petri dish, the new roots, which had been sterilized with 70% C_2_H_5_OH for 2 min and 1% NaClO for 20 min, were added uniformly around the new roots and incubated for 25 days. After 25 days of incubation, the roots were washed with pure water to remove ungerminated *O. aegyptiaca* seeds, and then the parasitism rate was determined at different periods (S4, S5, S6) [63,64]. The S4 period means that *O. aegyptiaca* produces spider web-like structures when the aspirator expands. The S5 period means that the pre-emergence shoot structure of *O. aegyptiaca* occurs. The S6 period means that there are post-emergence shoots of *O. aegyptiaca*.

Thirty treatments and ten controls were set up in three biological replicates.

### 4.4. Target Fragment Amplification

Total RNA was extracted from sampled leaves using the TRIzol method. Then, RNA was reverse transcribed to cDNA using a reverse transcription kit (*EasyScript*^®^ One-Step gDNA Removal and cDNA Synthesis SuperMix, Tran, FL, USA). The target sequence of the *SLJAZ15* gene was amplified with specific primers, forward 5′-ATGGGGTCATCGGAAAATATG-3′ and reverse 5′-CTAGAAATATTGCTCAGTTTTAACAA-3′, designed using Primer 5.0 software.

### 4.5. Agrobacterium-Mediated Virus Infection

*Agrobacterium tumefaciens* strain GV3101 carrying the TRV vector was cultured in an LB liquid medium (containing 50 mg/mL kanamycin and 100 mg/mL rifampicin) until the OD600 was 0.8–1.0. Transformed *Agrobacterium tumefaciens* cells were then centrifuged (4 °C, 4000 rpm, 10 min) and resuspended in a buffer (10 mM MES, 10 mM MgCl, 200 μM Acetosyingone) until the OD600 was 0.6–0.8. *Agrobacterium tumefaciens* cells containing TRV1 were mixed with TRV2-derived constructs or TRV2 empty vectors at a volume ratio of 1:1. The mixed bacterial liquid was allowed to stand for 3 h and was then injected into seedlings at the three-to-four leaf stage using a 1 mL syringe.

### 4.6. VIGS Vector Construction

PCR products and the TRV2 empty vector were individually digested with the restriction enzymes *EcoR* I and *BamH* I. The correct PCR products were recovered using a product recovery kit (Nanjing Novozymes Biotechnology Co., Ltd., DC301, Shihezi, China). The target fragment was ligated into the empty vector. The vector was heat stimulated into *Escherichia coli* DH5α cells according to the manufacturer’s instructions. A single positive clone was screened and cultured in liquid LB (containing 50 mg/mL kanamycin). The validated constructs, TRV-*SLJAZ15*, TRV-PDS, and TRV1:00, were then imported into *Agrobacterium* GV3101 using the electroshock transformation method. The validated sequences of TRV-*SLJAZ15* were as follows: F-primer (5′-3′), GTGAGTAAGGTTACCGAATTCGTGAAACAAATCCTCAAAAACCCATGAATCT; R-primer (5′-3′), CGTGAGCTCGGTACCGATCCTTCTTGAATCAATTTGTTACCAAAACTCACACCAG. When TRV-PDS transgenic plants showed obvious leaf whitening, *O. aegyptiaca* was co-cultured with other treatments. A schematic diagram of VIGS vector construction is shown below (Figure 7).

### 4.7. Quantitative Real-Time PCR Analysis

As soon as the seedlings infected with *A. tumefaciens* carrying the TRV-PDS vector showed visual photobleaching symptoms on the first-grown leaves, samples were collected from the *SLJAZ15*-silenced and control plants. RNA extraction and cDNA synthesis were performed as described above. qRT-PCR was performed using the iQ5 system. The real-time quantitative PCR mixture for each qRT-PCR consisted of 10 μL of 2 × PerfectStart^TM^ Green qPCR SuperMix, 0.4 μL of each forward and reverse primer, 1 μL of cDNA, and 8 μL of ddH_2_O in a 20 μL reaction. The reaction was performed as follows: 94 °C for 30 s, followed by 40 cycles of 94 °C for 5 s, 60 °C for 30 s, and 72 °C for 10 s; 3 biological replicates were included. The 2^−△△CT^ method was used for data analysis, and *ATActin* (Fw: 5′-ATGTCGACAACGGCTCCGGCATG-3′; Rv: 5′-GCTCGGGCACGACAGCACAGCTT-3′) was used as a reference gene. The sequences of gene-specific primers used were as follows: *SLJAZ15* Fw, 5′-TGAAGAAGCTGCGTTAGCGT-3′; *SLJAZ15* Rv, 5′-TCACCGGCGAACAATTACCA-3′. Three replicates were performed, and plants with more than a 50% reduction in gene expression were selected for subsequent *O. aegyptiaca* inoculation treatments.

### 4.8. Subcellular Localization of SLJAZ15

pCAMBIA2300-GFP is a subcellularly localized expression vector with a GFP tag applied in this experiment and kept by the laboratory. The pCAMBIA2300-*GFP* plasmid and full-length *SLJAZ15* were fused into subcellular localized vectors using *Sac* I and *Xba* I as the digestion sites and then transferred into DH5α-competent propagating cells using the thermal stimulation method. After obtaining the positive clone, it was transferred into *A. tumefaciens* to obtain the pCAMBIA2300-*SLJAZ15*-*GFP* vector. The specific primers used were Fw: 5′-AGAACACGGGGGACGAGCTCATGGGGTCATCGGAAAATATGGATTC-3′ and Rv: 5′-ACCATGGTGTCGACTCTAGAGAAATATTGCTCAGTTTTAACAAATTGAGCAC-3′. Transient transformation and laser confocal microscopic observation of tobacco leaves were carried out according to the method by Sparkes [25]. Additionally, we eversed three and four leaves, which were earmarked for injection (injection between two leaf veins). A syringe was pressed against the designated area on the posteriorside of the leaf, with one hand applying pressure on the top of the leaf, while the other hand gently depressed the piston until liquid diffusion was observed. Subsequently, following theinfestation, we circled the infected area using a marker, afterwards dousing the leaves with water and placing them in a plastic bag. Finally, we relocated the contaminated tobacco back to its original location. We opened the freshness bag the next day, cut the infested area 2–3 days after injection, tore the epidermal preparation, and observed the results using a fluorescence microscope. The green fluorescence of the GFP fusion protein (excitation wavelength of 488 nm) was observed and visualized with DAPI (excitation wavelength of 340 nm) using a FV10-ASW Laser Confocal Microscope (Leica, Wetzlar, Germany).

### 4.9. Observations of the Generation and Development of Heterologous Overexpression in Transgenic Plants

A fusion overexpression vector was constructed by inserting *SLJAZ15* CDS into a pCNY3 vector digested by *Xbal* I and *Sma* I. The vector was used for the construction of the fusion overexpression vector. In this study, *A. thaliana* was transgenic using the Floral Dip method [65], in which sequences from the T-DNA region of the vector were integrated into the DNA of *A. thaliana* using the *Arabidopsis* inflorescence infestation method, resulting in overexpression-positive *A. thaliana* plants. Kanamycin was used to screen the transgenic lines and was verified with genomic detection PCR. The *SLJAZ15* primers used for PCR were as follows: Fw, 5′-TCTAGAATGGGGTCATCGGA-3′; Rv, 5′-CCCGGGGAAATATTGCTC-3′. *SLJAZ15* high-expressing strains were screened and cultured to T2 generation, and T2 generation *SLJAZ15*-overexpressing seeds and Col 0 (WT) were simultaneously seeded in a plant culture room. After culturing to the appropriate stage, we parasitized *O. aegyptiaca* seeds that had been pretreated with GR24 to grow haustorium in Petri dishes and observed the parasitization results after 20 days of culture.

### 4.10. Physiological Measurements

Leaf samples were randomly collected from *SLJAZ15*-CK, *SLJAZ15*-T, WT-CK, and WT-T after *O. aegyptiaca* parasitization. Each sample included three replicates. Determination of the JA content in plants was performed using an ELISA Kit (Shanghai Yuanjie Biological Co., Shanghai, China).

### 4.11. Statistical Analysis

All the experiments were performed using at least three technical replicates or three biological replicates. The data analyses were performed using IBM SPSS Statistics 26 and GraphPad Prism software (v: 9.0.0.121). The differences between various groups were analyzed using the LSD test.

## Figures and Tables

**Figure 1 plants-13-01493-f001:**
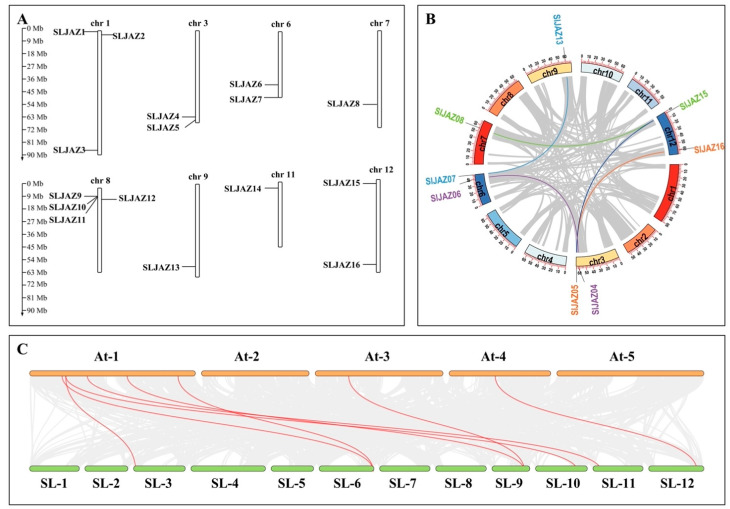
Chromosomal distribution of tomato *JAZ* genes and gene duplications of *JAZs*. (**A**) Chromosomal distribution of *SLJAZ genes*. Chromosome length can be judged from the left scale. (**B**) Gene duplication in *JAZs*. The inner circle shows the chromosome position where the gene is located. The same color indicates gene duplication. (**C**) Status of homologous *SLJAZ* genes in tomato and *A. thaliana*. Red line connections represent homology between the different species. The first line is the *A. thaliana* chromosome, and the second line is the *S. lycopersicum* chromosome. *SLJAZ1*, Solyc01g005440; *SLJAZ2*, Solyc01g009740; *SLJAZ3*, Solyc01g103595; *SLJAZ4*, Solyc03g118540; *SLJAZ5*, Solyc03g122190; *SLJAZ6*, Solyc06g068930; *SLJAZ7*, Solyc06g084120; *SLJAZ8*, Solyc07g042170; *SLJAZ9*, Solyc08g036660; *SLJAZ10*, Solyc08g036640; *SLJAZ11*, Solyc08g036620; *SLJAZ12*, Solyc08g036505; *SLJAZ13*, Solyc09g065630; *SLJAZ14*, Solyc11g011030; *SLJAZ15*, Solyc12g009220; *SLJAZ16*, Solyc12g049400.

**Figure 2 plants-13-01493-f002:**
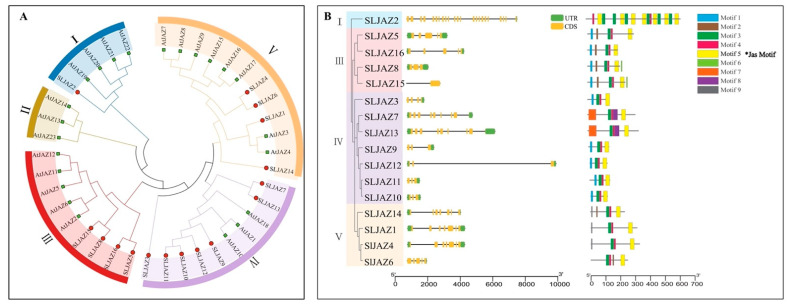
Phylogenetic tree of *JAZ* genes in tomato and *Arabidopsis thaliana* and gene structures of *SLJAZ* genes’ family members. (**A**) Phylogenetic tree of *SLJAZ* and *AtJAZ* genes. (**B**) Gene structure of *JAZ* gene family members in tomato. Branches of the same color represent the same subgroup. Asterisks indicate the motifs shared by *SLJAZ* genes.

**Figure 3 plants-13-01493-f003:**
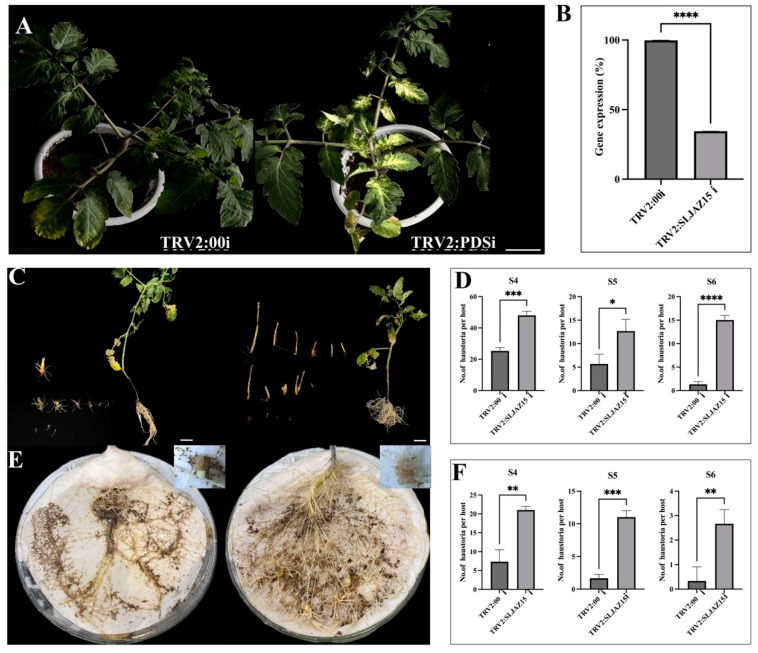
Efficiency of gene silencing and parasitic results of *O. aegyptiaca.* (**A**,**B**) Bleaching phenomenon in tomato plants and qRT-PCR results. (**C**,**D**) Co-cultivation of *O. aegyptiaca* with tomato for 20 days with the potting method. (**E**,**F**) Co-cultivation of *O. aegyptiaca* with tomato for 20 days with the root chamber method. (**A**) Bleaching of tomato plants after inoculation with the TRV2:00i (left) and TRV2:PDSi vector (right). (**B**) qRT-PCR results after *SLJAZ15* gene silencing. (**C**) *O. aegyptiaca* co-cultivated with TRV2:00i (left) and TRV2:*SLJAZ15*i (right) tomato for 20 days with the potting method. (**D**) Parasite differences between TRV2:00i and TRV2:*SLJAZ15*i co-cultured with *O. aegyptiaca* during S4–S6 for 20 days with the potting method. (**E**) *O. aegyptiaca* co-cultivated with TRV2:00i (left) and TRV2:*SLJAZ15*i (right) tomato for 20 days with the root chamber method. (**F**) Parasite differences between TRV2:00i and TRV2:*SLJAZ15*i co-cultured with *O. aegyptiaca* during S4–S6 for 20 days with the root chamber method. S4: *O. aegyptiaca* produces spider web-like structures when the aspirator expands. S5: Pre-emergence shoot structure of *O. aegyptiaca.* S6: Post-emergence shoots of *O. aegyptiaca*. * Represents *p* < 0.05, ** represents *p* < 0.01, *** represents *p* < 0.001, and **** represents *p* < 0.0001.

**Figure 4 plants-13-01493-f004:**
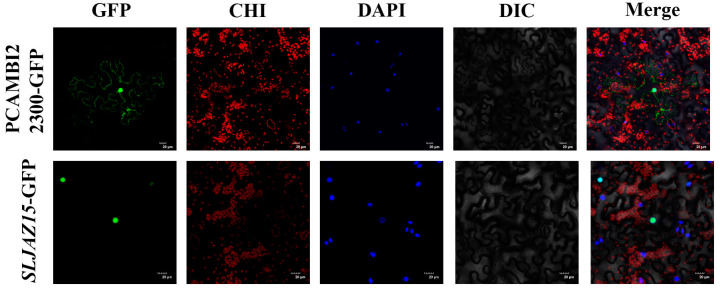
Subcellular localization of *SLJAZ15* in *Nicotiana tabacum.* GFP stands for green fluorescence field; DAPI stands for DAPI field; CHI stands for chloroplast autofluorescence field; DIC stands for bright field; and Merge stands for superimposed field. Excitation light wavelength: GFP field, 488 nm; DAPI field, 358 nm; and CHI field, 488 nm. Note that green fluorescence and chloroplast autofluorescence have the same excitation light but different acquisition light wavelengths. Bar = 20 μM.

**Figure 5 plants-13-01493-f005:**
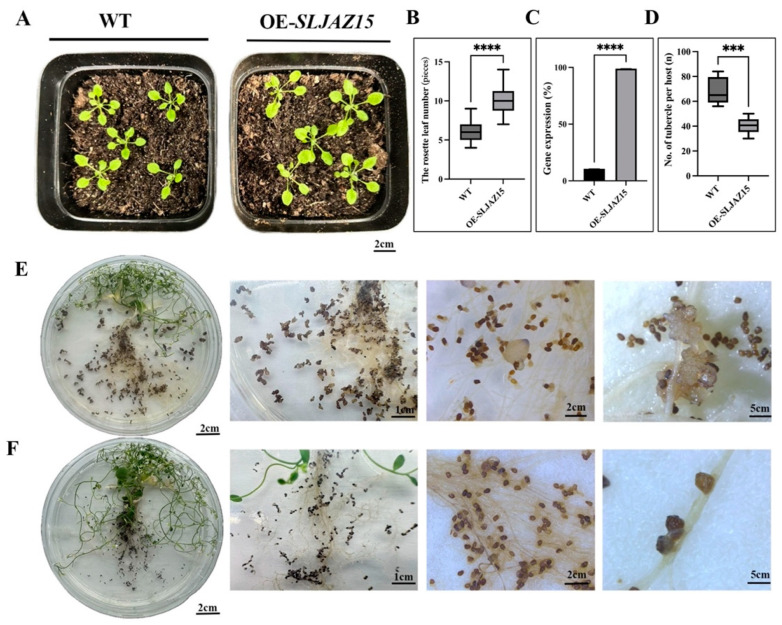
Heterologous overexpression of *SLJAZ15* in *Arabidopsis thaliana*. (**A**) Phenotypes of planting after *SLJAZ15* overexpression in *A. thaliana* after 10 d. (**B**) Data on the rosette leaf number in the wt and *SLJAZ15* overexpressing *Arabidopsis* plants after 10 days. (**C**) qRT-PCR results after *SLJAZ15* gene overexpression. (**D**) Number of tubercles per *A*. *thaliana* plant. (**E**) Parasitized phenotype of transgenic WT *A*. *thaliana* plants co-cultured with *O. aegyptiaca* for 20 days with the root chamber method. (**F**) Parasitized phenotype of transgenic *A*. *thaliana* plants OE-*SLJAZ15* co-cultured with *O. aegyptiaca* for 20 days with the root chamber method. *** Represents *p* < 0.001, and **** represents *p* < 0.0001.

**Figure 6 plants-13-01493-f006:**
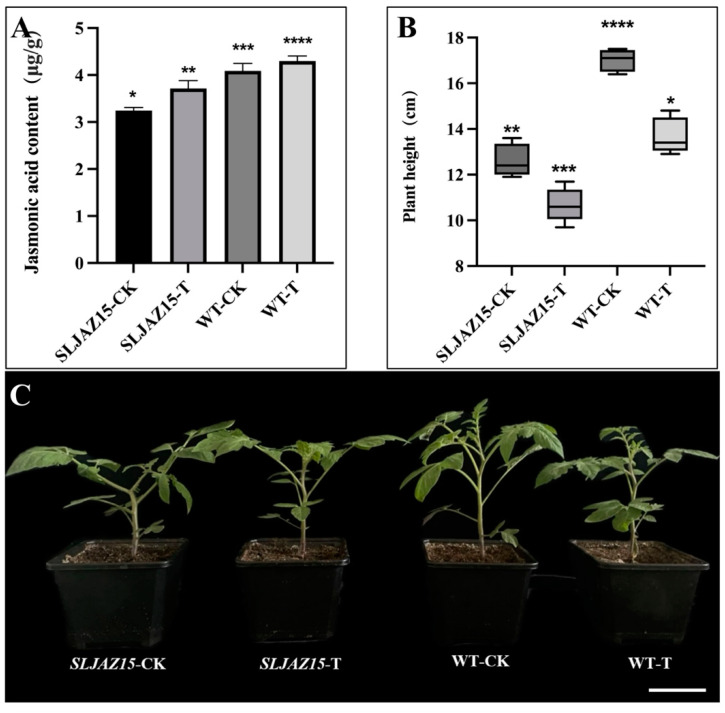
Results of physiological and biochemical analyses of *SLJAZ15* and morphological phenotype of transgenic tomato plants. (**A**) Jasmonic acid content in different tomato plants. (**B**,**C**) Height data and phenotypes of different tomato plants. (**B**) Height data for different tomato plants. (**C**) Phenotypes of different tomato plants. *SLJAZ15*-CK, *SLJAZ15-*silenced plants after 15 d; *SLJAZ15*-T, co-cultivation of *SLJAZ15-*silenced plants with *O. aegyptiaca* after 15 d; WT-CK, common tomato plants after 15 d; and WT-T, co-cultivation of common tomato plants with *O. aegyptiaca* after 15 d. * Represents *p* < 0.05, ** represents *p* < 0.01, *** represents *p* < 0.001, and **** represents *p* < 0.0001.

**Figure 7 plants-13-01493-f007:**
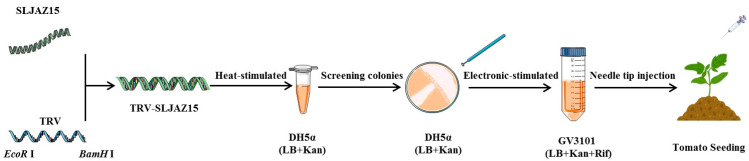
Schematic diagram of VIGS vector construction. *SLJAZ15*, target resistance genes; TRV, tobacco rattle virus; EcoR I, the restriction enzymes; BamH I, the restriction enzymes; TRV-*SLJAZ15*, vehicle for integration; DH5α (LB + Kan), DH5α cultured in an LB medium (with Kan); GV3101 (LB + Kan + Rif), cultured in an LB medium (with Kan and Rif).

## Data Availability

The data presented in this study are available on request from the corresponding author.

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
