# Peer review of "Jasmonate ZIM Domain Protein (JAZ) Gene SLJAZ15 Increases Resistance to Orobanche aegyptiaca in Tomato"

_plants, 2024, doi:10.3390/plants13111493_

Round 1
Reviewer 1 Report (Previous Reviewer 2)
Comments and Suggestions for Authors
Compared to the first version, the quality of figures have been improved and the text has been corrected and implemented in the missing parts as requested in the first revision step.
Some minor errors:
- There are some missing spaces in the text every now and then. Check again.
- Reference 13 is wrong: the exact title refers to Orobanche aegyptiaca while in the paper it is reported as Phelipanche aegyptiaca
- line 195 replace “flod” with “fold”
- Line 200: S4, S5 and S6: it’s better to explain the three periods also in the text, not only in the caption of figure 3
-in paragraph 2.5 (lines 244-247), the figure cited is not consistent with the text
-line272: JA content values are probably reversed
-Line 377: for a better understanding, specify that the text refers to tomato plants (given that in the paper you also talks about A. thaliana).
-in paragraph 3.4 conclusion, a sentence is missing on the phenomenon of dwarfism observed in SLJAZ15 silenced plants.
Comments on the Quality of English Languagejust a few typos
Author Response
Please see the attachment.

Reviewer 2 Report (Previous Reviewer 4)
Comments and Suggestions for Authors
FIG.1 – Nothing can be read. As reported in the previous revision, please increase the size of characters. THIS IS IMPORTANT. In an article, the figures are the most important part. Thus, it is important to design them in a way in which they are easily understood by the readers.
2.2 Phylogenetic relationships and gene structure analysis of SlJAZ genes
Line 153 – As reported in the previous revision, what do you mean by saying “23 AtJAZ proteins co-expressed in A. thaliana”? What I mean is that you should not use “co-expressed” because is misleading here, it is a specific scientific definition that should not be use in the simple grouping of genes. Moreover, since you are showing a Phylogenetic tree of genes, you should talk about genes.
Line 156 – 157 – Again, as asked in the previous revision, why SlJAZ15 is one of the most interesting genes? Please explain it, readers cannot know it if you do not do it. Everything should follow a logic, thus it is important to give to the readers the tools about why you have chosen to deepen the study of SLJAZ15.
FIG.2 – Same as Fig.1
2.4 Subcellular localization of SLJAZ15 in Nicothiana tabacum
Regarding Nicothiana, I was wrong during the first revision and you were right, the correct way to write it is “Nicotiana”. I apologize for the incorrect indication.
Fig.4, Line 235 – what do you mean by “cytosolic staining”? As you show in your picture, DAPI is a DNA-binding fluorophore. Still about this part, as pointed out in the first revision, in MeM section (line 542-544) the information reported are wrong. The excitation wavelength of DAPI is not 488 nm. For each channel you showed (GFP, CHI, DAPI) you must indicate: excitation wavelength, emission wavelength, objective that you use, laser power, pinhole aperture.
Moreover, you corrected the misleading sentence “subcellularly localized expression vector” from the 2.4 paragraph, but you left it in MeM as “subcellularly localized fusion vector” that is also wrong. Please correct, and if this type of sentence is present somewhere else in the text, please change it.
Regarding pCAMBIA2300-GFP and PCAMBIA2300-SLJAZ15-GFP, since we are talking about DNA they should be written entirely in italic. Moreover, as I asked before, what is the promoter that in planta drives the expression of these proteins?
2.5 SLJAZ15 overexpression in A. thaliana increased resistance to O. aegyptiaca
Line 240 – “To characterize the function of SLJAZ15 in tomato, A. thaliana was transformed…”… my previous review was pointing out that is quite counterintuitive that you want to characterize the function in tomato but you are going to work on Arabidopsis. You should write it in another way. As an example, you could write that you aim to investigate whether the function of SLJAZ15 is conserved.
Line 241 – “A. agalactiae”, do you mean A. tumefaciens?
Line 241-243 – please insert a citation about floral dip.
Line 258 – “Data on plant connecting leaves” is a strange sentence. I suggest to replace with “Data on the rosette leaf number in wt and SLJAZ15 overexpressing Arabidopsis plants”.
As I asked before in the previous revision, what is the promoter used to drive the overexpression of SLJAZ15 in Arabidopsis?
As asked before, I truly suggest you consider putting an “i” or something like this when you are talking about the silencing construct “TRV2:SLJAZ15”. Otherwise, it could be confused with the construct used for the overexpression. I suggest writing something like “TRV2:SLJAZ15i” (from RNAi), or TRV:SLJAZ15RNAi”, or “TRV:SLJAZ15-RNAi”.
In general, in all the figure I suggest to increase the character size.
Author Response
Please see the attachment.

Reviewer 3 Report (Previous Reviewer 5)
Comments and Suggestions for Authors
Authors addressed previous concerns only partially, with several major issues lacking correction. Thus, the work still not suitable for publication.
Additional comments can be found on the PDF file, and are presented below:
1. Authors often mix the methodology applied with the results obtained and their discussion, which makes it difficult to follow the text and the implications about the underlying mechanism.
2. The conclusion does not respond to a clear scientific question/hypothesis. Instead, it is only another abstract, summarizing the work. Thus, authors must draw a clear question/hypothesis in the end of the introduction, and also draw a conclusion in response to them.
3. Also, in the end of the abstract, authors state: “Finally, the function of the gene was further elucidated by using an enzyme activity assay and plant height measurement.”; but they do not provide this elucidation, which must be clearly and shortly presented in the end of the abstract and in conclusion.
4. Line 102-103: The correction during R1 was not addressed. Previous comment: "the most SlJAZ are NOT JA and JA-isoleucine mimic. They are induced by JA, abiotic stresses, and phytotoxins. Another issue is that this phrase is VERY similar to the citation last line of the abstract on reference [35]. Please rewrite"
5. Line 156-157: The correction during R1 was not addressed. Previous comment: “Why was SlJAZ15 stated as one of our most interesting genes (Line 153)? Please, provide some context in the text, justifying why did you choose this gene.”
6. Line 376-378: Comment not addressed: "this is part of the methodology and does not apply to the discussion section".
7. Plagiarism issues in the following lines were not removed:
- Lines 71-99: It is identical to Jia et al. 2021 (https://doi.org/10.1038/s41598-021-99593-2);
-Line 99-100: It is very similar to Chini et al. 2017 (https://doi.org/10.1371/journal.pone.0177381);
- Lines 436-438 (now lines 295-298 in the new version); 446-460 (now 305-319): It is identical to Sun et al. 2021 (https://doi.org/10.3390/ijms22189974).

Round 2
Reviewer 3 Report (Previous Reviewer 5)
Comments and Suggestions for Authors
Authors addressed the points previously raised. It is important to note that some plagiarism issues were solved only by changing few words, which is still ethycally problematic and, therefore, unacceptable. Please, rewrite the previously cited parts with your own words.
Please, provide proper citations in lines 64-66, 67-70.
Author Response
Please see the attachment.

This manuscript is a resubmission of an earlier submission. The following is a list of the peer review reports and author responses from that submission.
Round 1
Reviewer 1 Report
Comments and Suggestions for Authors
Comment 1: Write ‘Castanea mollissima’ instead of Castaneamollissima in second last paragraph of introduction section.
Comment 2: Font style should be uniform in whole the text.
Comment 3: Italicize the scientific name in last paragraph of introduction section and first line under material and methods section.
Comment 4: Rewrite the text “Virus-induced gene silencing ofGhCER04Ain upland cotton” under the material and method section.
Comment 5: Check the font style under subheading “Expression Profiling of GhCER Genes in response to Abiotic Stress”.
Reviewer 2 Report
Comments and Suggestions for Authors
This work has led to understanding the function of a key gene in tomato, a member of the JAZ gene family, involved in the host's resistance response to parasitic plants. Tomato plants with the silenced gene are much more susceptible to attack by the parasite, while when the gene is overexpressed (in Arabidopsis) the plant is more resistant to the attack.
The work is interesting and helps to better understand the mechanisms of interaction between plant and host. The conclusions are supported by the results.
However, the presentation of the work needs to be revised, the discussion is too descriptive while the introduction is too general and not focused on the true purpose of the research. English needs to be improved
Some corrections:
- the names of the genus and species are in italics,
- line 31 a dot to remove
- the SlJAZ gene name must be consistent throughout the text.
- Introduction: the purpose of the research explained in the introduction is too vague. It is not clear which key genes the work is focused on. Furthermore, in the first chapter of the results we immediately talk about SlJAZ genes without having ever been defined before (excluding the title) nor contextualized with bibliographical references in the introduction, while it is extensively (too!) described in the discussion; In my opinion the order should be reversed.
- Results: the description of the genetic transformation procedure for gene silencing, especially regarding the construction of transformation vectors, is scarce. Perhaps a figure could help describe the method.
- In paragraph 2.3 we talk about TRV-PDS but it is not explained what it is and what it is used for (a transformation control?)
- Figure 3: What do S4, S5 and S6 periods mean?
- Materials and methods lack a paragraph relating to statistical analysis
Comments on the Quality of English LanguageEnglish needs to be improved
Reviewer 3 Report
Comments and Suggestions for Authors
Please use the name Orobanche aegyptiaca Pers. instead of Phelipanche aegyptiaca, as the former is the accepted name. Also, please find the attached file.

Reviewer 4 Report
Comments and Suggestions for Authors
The work “Jasmonate ZIM-Domain protein (JAZ) Gene SlJAZ15 Increases Resistance to Phelipanche aegyptica in Tomato” by Chen and colleagues is potentially interesting and investigates new strategies to increase the resistance of tomato plants to infestation by Phelipanche aegyptiaca. Specifically, authors exploit the Virus-Induced Gene Silencing system (VIGS) to direct the silencing of SlJAZ15, following which they observed a lower P. aegypthica infection efficiency. Authors also characterize the role of SlJAZ15 by overexpressing it in Arabidopsis. Despite the potential interest, I find the work not well written, with gaps and scientific concepts not properly described. My comments are reported below. I advise authors to rewrite the work with more care and accuracy, and with attention to detail. Importantly, not being a bioinformatics expert I have not commented on the methodology in paragraphs 2.1 and 2.2.
Introduction
I found the introduction from line 46 to line 58 misleading, with a number of notions unnecessary to the work. I suggest the authors rather supplement with a better explanation of the VIGS system
Results
2.2 _ Phylogenetic relationships and gene structure analysis of SlJAZ genes
Line 150 _ what do you mean with “co-expressed”?
Line153-154 _ why is SlJAZ15 one of most interesting genes? Please explain briefly.
Line 160-161 _ Considering that I am not an expert of the field, as personal consideration I think that having the same number of exons is not a proof of homology but only an indication. Please rewrite the sentence in a softer way.
Fig. 2 _ In general, I suggest to enlarge a little the size of characters, since actually it is not possible to read nothing except for the groups number.
2.3 _ Resistant plants showed increased parasitism of P. aegyptiaca after silencing SlJAZ15
First of all, why did the authors decide to target SlJAZ15? Please explain
Line 174 _ Authors exploit the silencing of Phytoene desaturase (PDS) gene, which normally results in the bleaching of the sample, to validate the validity of the virus-induced gene silencing system. But this concept should not be considered taken for granted and authors should spend a few lines explaining it to the reader.
Line 178 _ I suggest inserting in the name of the construct any clue that this is a silencing construct. As an example authors could put a subscript “i” at the end of the name. Please consider this in all the work.
Line 183 – 186 _ Please refers to the figure, even if indicated in the subsequent sentence.
Line 188 _ What do the author mean with S4, S5 and S6 periods? Please explain.
Line 186-188 _ rewrite the sentence specifying that the parasitisation increased in plants in which the SlJAZ15 was silenced respect to plants carrying the TRV2 vector alone.
2.4 _ Subcellular location of SlJAZ15 in Nicotiana tabacum
First of all, it is better to talk about subcellular LOCALIZATION and not location. Moreover, Nicotiana should be written as “Nicothiana”.
Line 215-217 _ Please rewrite. What do you mean by talking about subcellularly localized expression vector? Moreover, a vector is not transformed but is Agrobacterium that is transformed by inserting the vector.
Line 217-220 _ what is PCAMBIA2300-GFP? Also in this case, this concept should not be considered taken for granted. Moreover, how is it possible that GFP without any type of tag remains in the membrane? The fact that the signal is around the cell periphery is due to the presence of the vacuole. Moreover, the description of how the images were acquired are reported wrongly in the “Material and methods” section, please rewrite. Moreover, for completeness of the experiment, the plants should also be inoculated with a specific nuclear marker and see if there is colocalization of the signal with SlJAZ15-GFP.
2.5 _ SlJAZ15 overexpression in A. thaliana increased resistance to P. aegyptiaca
Line 228 _ What do you mean by “characterizing the function in tomato by transforming Arabidopsis”?
Line 229 _ “inflorescence infiltration method”, please rewrite as Floral Dip, explaining briefly what it is.
Line 229-230 _ How can you claim that the SlJAZ15 is overexpressed in the generated transgenic lines? Please indicate the promoter under which SlJAZ15 is expressed, and secondly it is better to perform a qRT-PCR to validate the expression of the gene.
Line 230-233 _ I think that references to the figure reported in the text are wrong. Moreover, graphs reporting all the data cited in the text should be produced (same for other part of the work).
240-241 _ What do you mean by “normal plants”? Please refer to wt Arabidopsis plants and transgenic Arabidopsis plants expressing…
Fig. 5C What is the graph about?
Line 243 and others _ what do you mean by “stabilization of SlJAZ15 overexpression”?
Fig5 D-E _ data should be presented also with bar graphs and statistic.
Line260-264 – Please specify that we are in presence of P. aegyptiaca.
Reviewer 5 Report
Comments and Suggestions for Authors
1) The abstract must have up to 200 words.
2) The reference 15 cited on line 62 do not correspond to the information presented. Please, correct.
3) The reference 19 cited on line 63 do not correspond to Carica papaya and tomato intercropping. Moreover, it is necessary to stress the parasite species you are referring to. Please correct.
4) Add a reference to the information on lines 73-74.
5) Line 99: the most SlJAZ are NOT JA and JA-isoleucine mimic. They are induced by JA, abiotic stresses, and phytotoxins. Another issue is that this phrase is VERY similar to the citation last line of the abstract on reference [35]. Please rewrite.
6) Why was SlJAZ15 stated as one of our most interesting genes (Line 153)? Please, provide some context in the text, justifying why did you choose this gene.
7) The font size used in the figure 2 is still very small, mainly on Figure 2A. Please improve it.
8) Is the tomato variety “H1015” resistant to P. aegyptiaca? If so, please provide some context in the introduction and “3.3 Plant material and treatments” sections.
9) Please, specify if the “TRV2” plants on line 198 are the TRV2:00 plants.
10) The Figure 3A shows necrosis in the TRV2:PDS; however, in the Figure 3C, the necrotic plant (left) is called as TRV2:00, which generates confusion. Please, revise the information on figure 3 and on the text, being consistent on how the transformed plants are called (:PDS or :SLJAZ15) and the position they are placed on the figure (left or right). For this, place the TRV2:00 and the TRV2:PDS always in the same position (TRV2:00 on the left, and TRV2:PDS on the right, for example). If possible, please also include the identification on the Figure 3C.
11) Please, add a graph or table showing the results for average number of P. aegyptiaca parasitizing Arabidopsis plants (cited on lines 233-235).
12) Why is the “Methods” section placed between results and discussion? I suggest to place it before results or after discussion, as well as calling it as “Materials and methods”, following the journal guidelines.
13) Line 495: this is part of the methodology and does not apply to the discussion section.
14) Line 509: same as above. It is not necessary to keep repeating the methodology in the discussion. The same occurs on lines 521-523.
15) Please, provide a short conclusion either at the end of the discussion, or as a separate section, also adding this to the abstract.
16) Regarding the language, some typos and writing suggestions across the manuscript were highlighted in yellow in the PDF version and are listed below:
a. Line 9: First line of abstract: there is an additional “is” after the species name.
b. Line 23: Same as above.
c. Line 31: Suggestion: “P. aegyptiaca is an increasing threat to tomato cultivation, as they parasitize on plant roots leading to devastating yield losses”.
d. Line 46: The phrase about cucumber expansins lacks context. I suggest rewriting it, maybe starting it with “for example”, and removing the “for example” from the next phrase; or placing it after the phrase that starts with “for example”.
e. Line 98: What do you mean with “oral” organ development? Revise and correct it, specifying which JAZ you are referring to.
f. The text on lines 99-111 requires English language revision by a native speaker.
g. Figure 3B and 5C: please correct the y axe title to “gene expression”. Also, specify, on the figure or in the caption, that the relative expression evaluated is respective to the SlJAZ15 gene.
Comments on the Quality of English Language16) Regarding the language, some typos and writing suggestions across the manuscript were highlighted in yellow in the PDF version and are listed below:
a. Line 9: First line of abstract: there is an additional “is” after the species name.
b. Line 23: Same as above.
c. Line 31: Suggestion: “P. aegyptiaca is an increasing threat to tomato cultivation, as they parasitize on plant roots leading to devastating yield losses”.
d. Line 46: The phrase about cucumber expansins lacks context. I suggest rewriting it, maybe starting it with “for example”, and removing the “for example” from the next phrase; or placing it after the phrase that starts with “for example”.
e. Line 98: What do you mean with “oral” organ development? Revise and correct it, specifying which JAZ you are referring to.
f. The text on lines 99-111 requires English language revision by a native speaker.
g. Figure 3B and 5C: please correct the y axe title to “gene expression”. Also, specify, on the figure or in the caption, that the relative expression evaluated is respective to the SlJAZ15 gene.
